# Estimation of Pesticide Residues in Selected Products of Plant Origin from Poland with the Use of the HPLC-MS/MS Technique

**Grażyna Kowalska [1], Urszula Pankiewicz [2] and Radosław Kowalski [2,*]**

[1] Department of Tourism and Recreation, University of Life Sciences in Lublin, 15 Akademicka Street, 20-950 Lublin, Poland; grazyna.kowalska@up.lublin.pl

[2] Department of Analysis and Evaluation of Food Quality, University of Life Sciences in Lublin, 8 Skromna Street, 20-704 Lublin, Poland; urszula.pankiewicz@up.lublin.pl

\* Correspondence: radoslaw.kowalski@up.lublin.pl

**Abstract:** The purpose of this work was to compare the content of pesticide residues (250) in unprocessed plant products from farms situated in the eastern part of Poland. The content of pesticide residues in the analysed samples was assayed with the use of the QuEChERS (Quick Easy Cheap Effective Rugged Safe) method combined with HPLC-MS/MS (high performance liquid chromatography with tandem mass spectrometry) analysis. The analyses revealed that among 160 analysed samples, pesticide residues were detected in 83 samples (approximately 52%), while in 77 samples (approximately 48%), no presence of those substances was noted. In all the samples in which the presence of the sought compounds was identified, their levels did not exceed the Maximum Residue Levels (MRL). The most often identified ones were azoxystrobin—detected in 36 samples (22.5%), linuron—assayed in 33 samples (20.6%), chlorpyrifos and carbendazim—each detected in 13 samples (8.1%), metalaxyl and metalaxyl M—in 11 samples (6.9%), and acetamiprid—in 7 samples (4.4%).

**Keywords:** pesticide residues; QuEChERS; LC-MS/MS; vegetables; fruits; herbs; spices

## 1. Introduction

The estimation of contaminants and chemical residues in food of plant origin assumes a significant importance, which is related with the progress of science and results from the critical attitude of consumers towards the applied methods of agricultural production and to the environmental pollution [1]. Accumulation of pesticide residues in food may cause toxic and allergic effects for human health and life as a result of the consumption of contaminated products [2]. For the protection of public health, the European Union introduced the highest permissible levels of pesticide residues in food and feed of plant and animal origin, regulated by the Regulation (EC) No. 396/2005 of the European Parliament and Council on 23 February 2005. Quantitative assays of pesticide residues in food allow the estimation of the exposure of consumers to the presence of those compounds and to perform risk assessment. The results of such analyses also provide important information on actual levels of pesticide residues and may cause a modification of the scope of their application in agriculture for the purpose of reduction of excessive levels relative to the Maximum Residue Levels (MRL). A highly important aspect in the estimation of the presence of pesticide residues is the application of a suitable analytical procedure that should meet the assumed requirements and guarantee the obtainment of results which can constitute the basis for making correct administrative decisions. Current studies in the field of estimation of pesticide residues indicate the universal character of the technique of liquid

chromatography with mass spectrometry (LC/MS/MS) in the analysis of that group of substances in samples of plant raw materials and in ready food products. Literature data confirm that the LC/MS/MS technique is characterised by adequate selectivity and specificity and allows to acquire, in the course of the analytical process, the required values of parameters confirming the quality of the result [3–10].

The quality requirements relating to food impose on the producers the necessity of controlling the quality of market products. Such a control results in an improvement of the quality of the food produced. One can also observe a trend towards minimisation of the number of plant protection treatments, but in spite of the existing legal regulations in this area, there are instances of breaking the regulations, resulting in the risk of products with exceeded limit levels for pesticide residues finding their way onto the market. In view of the above, the objective of this study was to compare the content of pesticide residues in 6 kinds of food products, i.e., vegetables, fruits, herbs, spices, and fruit and vegetable juices, as well as industrial plants originating from production farms in the eastern part of Poland.

## 2. Materials and Methods

### 2.1. Experimental Material

The research material consisted of samples of unprocessed plant products collected at random from farms situated in the eastern part of Poland, in the period of 2015–2016. Imported spices and juices were purchased in Lublin supermarkets. Minimum weight of a sample was 3 kg. The total number of samples was 160, classified into 6 groups:

1.  Vegetables (20)—carrot (2), cabbage (1), beetroot (2), root celery (1), parsley (2), green pea (1), cucumber (1), broccoli (1), pumpkin (3), beans (1), radish (1), chive (1), dill (1), peppers (1), field pea (1).
2.  Fruits (26)—blackcurrant (9), cherry (2), strawberry (4), blueberry (1), aronia berry (1), apple (3), pear (2), raspberry (2), elderberry (2).
3.  Herbs (85)—root of valerian (2), herbage of thyme (39), leaf of mint (3), root of common dandelion (4), leaf of lemon balm (3), herbage of common origanum (1), herbage of marjoram (1), fruit of coriander (3), linseed (17), leaf of small plantain (3), leaf of sage (1), herbage of rock rose (1), leaf of nettle (1) root of liquorice (1), flower of marigold (1), flower of elderberry (1), leaf of blackcurrant (2), leaf of purple coneflower (1).
4.  Spices (22)—black pepper (4), bay leaf (1), orange skin (1), fruit of caraway (3), curcuma (1), nutmeg (1), allspice (1), ginger (1), herbal spice (4), herbal pepper substitute (3), Herbes de Provence (2).
5.  Fruit and vegetable juices (4)—multifruit juice (1), pear juice (1), apple juice (1), beetroot juice (1).
6.  Industrial plants (3)—wheat (2), rape (1).

### 2.2. Chemicals

High-purity pesticide standards (250) were used for testing (98–99%, Dr. Ehrenstorfer GmbH, Augsburg, Niemcy; ChemService, West Chester, PA, USA): 2,4,5-T, 2,4-D, 2,4-DB, 3,5-Dichloroaniline, 3-hydroxycarbofuran, Abamectin, Acephate, Acetamiprid, Acrinathrin, Alachlor, Aldicarb, Aldicarb Sulfoxide, Aldicarb Sulphone, Ametryn, Amitraz, Atrazine, Azinophos-Ethyl, Azinophos-Methyl, Azoxystrobin, Benfuracarb, Bentazon, Benzoylprop ethyl, Bifenazate, Bromacil, Bromoxynil, Bromuconazole, Buprofezine, Butoxycarboxin, CAP (Captan), Carbaryl, Carbendazim, Carbetamide, Carbofuran, Carbosulfan, Carboxin, Chlorantraniliprole, Chloridazon, Chlorotoluron, Chlorpyrifos, Chlorsulfuron, Clofentezine, Clomazone, Clothianidin, Coumaphos, Cyanazine, Cyanofenphos, Cycloate, Cymoxanil, Cyphenothrin, Cyprofuram, DEF (Decafentin), Demeton-S-methyl, Demeton-S-methylsulphon, Desethyl atrazin, Desisopropyl atrazin, Desmedipham, Desmetryn, Diafenthiuron, Dialifos, Diazinon, Dicamba, Dichlofluanid, Dichloprop

(2.4-DP), Diclorvos, Dicrotophos, Diflubenzuron, Dimefuron, Dimethachlor, Dimethenamide, Dimethoate, Dimethomorph, Diniconazole, Diphenamide, Diphenylamine, Disulfoton, Ditalimfos, Diuron, DMF (2,4-Dimethyl-phenyl-formamidine), Dodine, Epoxiconazole, Etaconazole, Ethiofencarb, Ethirimol, Ethofenprox, Etoxazole, Etrimphos, Fenamidon, Fenamiphos, Fenazaquin, Fenbuconazole, Fenhexamid, Fenoxap-p-ethyl, Fenoxycarb, Fenpropimorph, Fenpyroximate, Fenthion, Fenthion sulfon, Fenuron, Fipronil, Flazasulfuron, Florosulam, Fluazifop, Fluazifop-p-butyl, Fluazinam, Fludioxonil, Flufenacet, Flufenoxuron, Fluometuron, Fluroxypyr, Flurtamon, Fluthiacet methyl, Flutriafol, Fonofos, Fosthiazate, Fuberidazol, Furathiocarb, Halfenprox, Haloxyfop, Haloxyfop methyl, Haloxyfop-2-ethoxyethyl, Heptenophos, Hexaflumuron, Hexazinone, Hexythiazox, Imazalil, Imazamox, Imazapyr, Imidacloprid, Indoxacarb, Ioxynil, Iprodione, Iprovalicarb, Isazofos, Isocarbamide, Isomethiozin, Isoproturon, Isoxaflutole, Lenacil, Linuron, Lufenuron, Malaoxon, Malathion, MCPA (2-Methyl-4-chlorophenoxyacetic acid), MCPB (4-(2-Methyl-4-chlorophenoxy) butyric acid), MCPP (Mecoprop), Mecarbam, Mepanipyrim, Metalaxyl, Metalaxyl-M, Metamitron, Metazachlor, Metconazol, Methabenzthiazuron, Methacrifos, Methamidophos, Methidathion, Methiocarb, Methoprotryne, Methoxyfenozide, Metobromuron, Metolachlor, Metolachlor S, Metosulam, Metoxuron, Metrafenon, Monocrotophos, Monolinuron, Monuron, Myclobutanil, Nicosulfuron, Nitenpyram, Norflurazon, Novaluron, Omethoate, Oxamyl, Oxycarboxin, Oxydemethon methyl, Paraoxon ethyl, Paraoxon methyl, Parathion ethyl, Pebulat, Penconazole, Pencycuron, Phenkapton, Phenmedipham, Phenothrin, Phenthoate, Phorate, Phosalone, Phosmet, Phosphamidon, Phoxim, Picoxystrobin, Pirimicarb, Pirimiphos methyl, Prochloraz, Profenofos, Prometryn, Propamocarb, Propanil, Propaquizafop, Prophos, Prosulfuron, Pyraclostrobin, Pyraflufen ethyl, Pyridaphenthion, Pyridate, Pyrimiphos ethyl, Pyriproxyfen, Quinmerac, Quizalofop-p-ethyl, Resmethrine, Rimsulfuron, Sebuthylazin, Sethoxydim, Siltiopham, Simazine, Simetryn, Spinosad A, Spinosad D, Spirotetramat, Spiroxamin, Sulfotep, Sulprofos, Tebuconazole, Tebufenozide, Tebufenpyrad, Tebutam, Teflubenzuron, Tepraloxydim, Terbucarb, Terbumeton, Terbuthialzine desethyl, Terbuthylazine, Tetramethrin, Thiabendazole, Thiacloprid, Thiamethoxam, Thiodicarb, Thiophanate methyl, Tolclofos methyl, Tolylfluanid, Triadimefon, Tri-allate, Triamiphos, Triazophos, Trichlorofon, Triclopyr, Trifloxystrobin, Triflumuron, Triforine. Standard solutions of pesticide in acetonitrile, with concentration of approximately 1000 mg L$^{-1}$, were prepared. Next, standard solutions of a mixture of pesticides in acetonitrile, with concentration of about 35 mg L$^{-1}$, were prepared for each of the compounds. Working standard solutions were prepared by diluting the standard mixtures of pesticide solutions with acetonitrile. All standard solutions were stored at temperatures lower than −20 °C. The choice of analysed pesticides resulted from the demand of herb producers' customers for analyses in line with the laboratory services market in the region. In addition, only pesticides for which the criteria for analytical quality were met were included in the analysis.

### 2.3. Preparation of Samples

The analytical procedure was described in earlier work [11]. Portions of about 3 kg of plant material were suitably mixed to obtained uniform material, and then samples of approximately 100 g were collected and homogenised. The obtained homogenisate was transferred in suitable amounts to 50 mL test tubes. In the case of dry matrices, the samples were moistened to the level of about 95%.

The next step was the addition, to the homogenisate, of 10 mL of acetonitrile (Merck, Darmstadt, Germany) and 100 μL of internal standard of triphenylphosphate (Merck, Darmstadt, Germany) (10 μg mL$^{-1}$) assayed in the mode of positive ionisation and 100 μL of internal standard of bis-nitrophenyl urea (Merck) (10 μg mL$^{-1}$) assayed in the mode of negative ionisation as an internal standard. The test tube was shaken vigorously for 1 min. Next, a mixture of salts QuECheRS Mix I (Agilent Technologies, Santa Clara, CA, USA) was added, and the tube was shaken again for 1 min and centrifuged for 5 min (1361 rcf). The obtained extract was purified by adding the mixture of salts QuEChERS Mix II (Agilent Technologies, Santa Clara, CA, USA), while in the case of samples containing chlorophyll, the mixture QuEChERS Mix III (Agilent Technologies, Santa Clara, CA, USA)

was additionally added, and the tube was shaken again for 1 min, and then centrifuged for 5 min (1361 rcf). The extract prepared in this manner was transferred to the autosampler vial and subjected to chromatographic analysis.

*2.4. Pesticides Analysis*

The content of pesticide residues in the analysed samples was assayed following a modified procedure developed in accordance with the standard PN-EN 15662:2008 [12], with the use of the method QuEChERS combined with LC-MS/MS analysis. The procedure applied in the study has been approved by the Polish Centre of Accreditation (PCA 1375).

HPLC MS/MS Analysis

A Shimadzu Prominence/20 series HPLC system (Shimadzu, Tokyo, Japan) and AB SCIEX 4000 QTRAP®LC-MS/MS system with Turbo V source (Foster City, California, USA) were used for LC-MS/MS analysis. The HPLC system was equipped with a LC-20 AD binary pump, a SIL-20 AC autosampler, a DGU-20A5 online degasser and a CTO-20A column oven. Nitrogen with a purity of at least 99% generated from a Peak Scientific nitro en generator (Billerica, MA, USA) was used in the ESI source and the collision cell. Analysis was performed using a $4.6 \times 100$ mm $\times 5$ μm Agilent ZORBAX Eclipse XDB C18 column with a 10 μL injection. The column temperature was constant at 40 °C. A mobile phase gradient of water with 5 mM ammonium acetate and methanol with 5 mM ammonium formate and flow rate of 0.5 mL min$^{-1}$ were used. Mobile phase was composed of HPLC-grade water containing 5 mM ammonium acetate (eluent A) and HPLC-grade methanol containing 5 mM ammonium acetate (eluent B). The gradient elution was performed as follows: 0–0.1 min: 20% B; 0.1–1 min: 20–45% B; 1–9 min: 45–80% B; 9–19 min: 80–100% B, 19–20 min: 100% B; 20–21 min: 100–20% B; 21–24 min: 20% B. A flow rate of 0.5 mL min$^{-1}$ and an injection volume of 15 mL were used in the LC-MS/MS system.

The mass spectrometer was operated using an ESI source in the positive and negative mode. ESI parameters were as follows: ion spray voltage 5.5 kV (ESI+) and −4.5 kV (ESI−), source temperature 600 °C, curtain gas (nitrogen) 35 psi, ion source gas "1" 50 psi, ion source gas "2" 65 psi, and collision gas (nitrogen) 5 psi. ESI-MS/MS was operated in scheduled multiple reaction monitoring mode (MRM), in both positive and negative polarities, by scanning two precursor/products ion transitions for each target analyte. Both transitions were used for quantification and confirmation purposes (see the Supplementary Material: Tables S1 and S2).

The recovery for pesticides in the matrices tested ranged from 70% to 120%. The limit criterion for linearity was the range above $r \geq 0.995$ (values from 0.9950 to 0.9998 were obtained).

## 3. Results

The analyses revealed that among 160 analysed samples, pesticide residues were detected in 83 samples (approximately 52%), while in 77 samples (approximately 48%), no presence of those substances was noted. In all the samples in which the presence of the sought compounds was identified, their levels did not exceed the Maximum Residue Levels (MRL). The occurrence of the analysed contaminants in the particular kinds of analysed samples is presented in Table 1. Residues of plant protection agents were found most often in samples of fruits—approximately 70%, while in herbs and fruit juices, pesticides were noted in approximately 53% and 50% of the samples, respectively. The lowest share of samples containing that group of analysed contaminants was noted in the case of vegetables—40%, and spices—approximately 43% (Table 1). Among the food samples subjected to analysis, pesticide residues were most frequently detected: in the group of herbs—in thyme (80%), in the group of fruits—in blackcurrant (44.4%), and in the group of spices—in black pepper (44.4%) (Table 2). Residues of two or more pesticides were hound in 54 samples (65.1%). In total, the presence of two pesticides was found in 25 samples (30.12%), the presence of three pesticides was noted in 11 samples (13.3%), and the presence of four and five pesticides, in 8 and 6 samples, respectively

(9.6% and 7.2%). One each of the analysed samples contained combinations of 7, 8, 9, and 12 of the identified compounds (Table 2). Co-occurrence of pesticide residues was noted in 44 herbal samples (91.8%), in 4 fruit samples (14.8%), in 2 vegetable samples (10%), and in 2 spice samples (9.5%). In the case of the herbal samples, the most often detected combination was that of a fungicide and a herbicide (azoxystrobin and linuron)—28 samples (32.9%), a combination of 2 fungicides with a herbicide (azoxystrobin, carbendazim and linuron) was assayed in 8 samples (9.4%), and combinations of 2 fungicides with 2 herbicides (azoxystrobin, linuron, metalaxyl, and metalaxyl M) were found in 8 samples (9.4%).

The presence of residues of an insecticide (acetamiprid) and a fungicide (trifloxysrobin) was found in 4 samples of fruits, in 2 samples of vegetables, a combination of a fungicide (azoxystrobin) and a herbicide (linuron) was detected, and the occurrence of a fungicide (azoxystrobin) and a herbicide (linuron) was noted in 2 samples of spices. In individual samples of herbs, the most often detected pesticide residues were linuron and azoxysrobin, in fruit samples—thiacloprid and trifloxystrobin, in spice samples—metalaxyl, metalaxyl M, and chloropyrifos, while in vegetable samples—azoxystrobin and chlorpyrifos (Table 2).

In the analysed samples, a total of residues of 40 pesticides were identified. The most often identified ones were azoxystrobin—detected in 36 samples (22.5%), linuron—assayed in 33 samples (20.6%), chlorpyrifos and carbendazim—each detected in 13 samples (8.1%), metalaxyl and metalaxyl M—in 11 samples (6.9%), and acetamiprid—in 7 samples (4.4%). The frequency of occurrence of all identified pesticides is presented in Figure 1. From among the 250 compounds sought in the presented experiment, in the analysed samples, the presence of 40 pesticides was found, which means that no presence of 210 pesticides from the estimated group of plant protection agents was detected. In terms of the use of the marked substances, they were classified into groups: fungicides (47.5%), insecticides (32.5%), herbicides (15%), carbamates (2.5%), and organophosphorus pesticides (2.5%). In the presented research, all identified pesticide residues are authorised in Poland. All pesticides found in individual products of plant origin are dedicated to the protection of a given plant species.

**Table 1.** Number of samples with and without detected pesticides residues for each analysed food product.

| | Food Product | | | | | | | | | | | | |
| | Vegetables | | Fruits | | Herbs | | Spices | | Fruit and Vegetable Juices | | Cereals | | Total | |
| | Number of Sumples | % | Number of Sumples | % | Number of Sumples | % | Number of Sumples | % | Number of Sumples | % | Number of Sumples | % | Number of Sumples | % |
|---|---|---|---|---|---|---|---|---|---|---|---|---|---|---|
| Samples analysed | 20 | - | 27 | - | 85 | - | 21 | - | 4 | - | 3 | - | 160 | - |
| No residues found | 12 | 60 | 8 | 29.6 | 40 | 47.1 | 12 | 57.1 | 2 | 50 | 3 | 100 | 77 | 48.1 |
| Residues found < MRL | 8 | 40 | 19 | 70.4 | 45 | 52.9 | 9 | 42.9 | 2 | 50 | 0 | 0 | 83 | 51.9 |
| Residues found > MRL | 0 | 0 | 0 | 0 | 0 | 0 | 0 | 0 | 0 | 0 | 0 | 0 | 0 | 0 |

MRL—Maximum Residue Levels.

**Table 2.** Pesticide residues concentration in examined food samples.

| No. | Food Product | Pesticide Residue | MRL (mg kg⁻¹) | LOQ (mg kg⁻¹) | Concentration (mg kg⁻¹) | Uncertainty (mg kg⁻¹) |
|---|---|---|---|---|---|---|
| | | Herbs | | | | |
| 1 | Thyme herb | Acetamiprid | 3.0 | 0.0001 | 0.026 | ±0.005 |
| | | Azoxystrobin | 70.0 | 0.0001 | 0.073 | ±0.026 |
| | | Carbendazim | 0.1 | 0.0001 | 0.052 | ±0.016 |
| | | Chlorpyriphos | 0.05 | 0.0001 | 0.012 | ±0.003 |
| | | Dimethoate | 0.02 | 0.0001 | 0.010 | ±0.003 |
| | | Linuron | 1.0 | 0.0002 | 0.014 | ±0.005 |
| | | Metalaxyl | 2.0 | 0.0001 | 0.046 | ±0.009 |
| 2 | Thyme herb | Azoxystrobin | 70 | 0.005 | 0.023 | ±0.008 |
| 3 | Thyme herb | Acetamiprid | 3.0 | 0.0001 | 0.018 | ±0.004 |
| | | Azoxystrobin | 70.0 | 0.0001 | 0.052 | ±0.018 |
| | | Carbendazim | 0.1 | 0.0001 | 0.027 | ±0.008 |
| | | Linuron | 1.0 | 0.0002 | 0.015 | ±0.005 |
| 4 | Thyme herb | Azoxystrobin | 70 | 0.0001 | 0.035 | ±0.012 |
| | | Carbendazim | 0.1 | 0.0001 | 0.042 | ±0.013 |
| | | Linuron | 1.0 | 0.0002 | 0.009 | ±0.012 |
| | | Metalaxyl | 2.0 | 0.0001 | 0.013 | ±0.003 |
| | | Metazachlor | 0.3 | 0.0001 | 0.024 | ±0.006 |
| 5 | Thyme herb | Azoxystrobin | 70.0 | 0.005 | 0.069 | ±0.024 |
| | | Linuron | 1.0 | 0.005 | 0.026 | ±0.008 |
| 6 | Thyme herb | Linuron | 1.0 | 0.005 | 0.057 | ±0.018 |
| 7 | Thyme herb | Azoxystrobin | 70.0 | 0.005 | 0.036 | ±0.017 |
| | | Linuron | 1.0 | 0.005 | 0.031 | ±0.012 |
| 8 | Thyme herb | Azoxystrobin | 70.0 | 0.005 | 0.098 | ±0.034 |
| | | Linuron | 1.0 | 0.005 | 0.022 | ±0.009 |
| | | Metalaxyl | 2.0 * | 0.002 | 0.028 | ±0.006 |
| | | Metalaksyl M | | 0.002 | 0.027 | ±0.005 |
| 9 | Thyme herb | Azoxystrobin | 70.0 | 0.005 | 0.013 | ±0.005 |
| | | Carbendazim | 0.1 | 0.002 | 0.08 | ±0.034 |
| | | Linuron | 1.0 | 0.005 | 0.032 | ±0.012 |
| 10 | Thyme herb | Azoxystrobin | 70.0 | 0.005 | 0.028 | ±0.010 |
| | | Carbendazim | 0.1 | 0.002 | 0.093 | ±0.035 |
| | | Linuron | 1.0 | 0.005 | 0.019 | ±0.007 |
| 11 | Thyme herb | Azoxystrobin | 70.0 | 0.005 | 0.042 | ±0.02 |
| | | Carbendazim | 0.1 | 0.002 | 0.022 | ±0.007 |
| | | Linuron | 1.0 | 0.005 | 0.027 | ±0.01 |
| | | Pyraclostrobin | 2.0 | 0.002 | 0.022 | ±0.007 |
| 12 | Thyme herb | Azoxystrobin | 70.0 | 0.005 | 0.009 | ±0.002 |
| | | Chlorantraniliprole | 20.0 | 0.005 | 0.270 | ±0.130 |
| | | Dimethoate | 0.02 | 0.002 | 0.140 | ±0.040 |
| | | Linuron | 1.0 | 0.005 | 0.012 | ±0.004 |
| 13 | Thyme herb | Azoxystrobin | 70.0 | 0.005 | 0.007 | ±0.001 |
| | | Linuron | 1.0 | 0.005 | 0.03 | ±0.010 |
| 14 | Thyme herb | Azoxystrobin | 70.0 | 0.005 | 0.053 | ±0.018 |
| | | Carbendazim | 0.1 | 0.002 | 0.021 | ±0.007 |
| 15 | Thyme herb | Metalaxyl | 2.0 * | 0.002 | 0.073 | ±0.015 |
| | | Metalaxyl-M | | 0.002 | 0.073 | ±0.015 |
| 16 | Thyme herb | Azoxystrobin | 70.0 | 0.005 | 0.031 | ±0.015 |
| | | Carbendazim | 0.1 | 0.002 | 0.086 | ±0.033 |
| | | Linuron | 1.0 | 0.005 | 0.029 | ±0.009 |
| | | Metalaxyl | 2.0 * | 0.002 | 0.018 | ±0.004 |
| | | Metalaxyl-M | | 0.002 | 0.018 | ±0.004 |

**Table 2.** *Cont.*

| No. | Food Product | Pesticide Residue | MRL (mg kg$^{-1}$) | LOQ (mg kg$^{-1}$) | Concentration (mg kg$^{-1}$) | Uncertainty (mg kg$^{-1}$) |
|---|---|---|---|---|---|---|
| 17 | Thyme herb | Chlorantraniliprole | 20.0 | 0.005 | 0.17 | ±0.080 |
| | | Linuron | 1.0 | 0.005 | 0.061 | ±0.019 |
| 18 | Thyme herb | Carbendazim | 0.1 | 0.002 | 0.062 | ±0.019 |
| | | Chlorantraniliprole | 20.0 | 0.005 | 0.160 | ±0.07 |
| 19 | Thyme herb | Carbendazim | 0.1 | 0.002 | 1.29 | ±0.005 |
| | | Linuron | 1.0 | 0.005 | 0.016 | ±0.005 |
| | | Metalaxyl | 2.0 * | 0.002 | 0.008 | ±0.002 |
| | | Metalaxyl-M | | 0.002 | 0.006 | ±0.002 |
| 20 | Thyme herb | Azoxystrobin | 70.0 | 0.005 | 0.067 | ±0.013 |
| | | Linuron | 1.0 | 0.005 | 0.110 | ±0.030 |
| 21 | Thyme herb | Azoxystrobin | 70.0 | 0.005 | 0.035 | ±0.007 |
| | | Linuron | 1.0 | 0.005 | 0.120 | ±0.040 |
| 22 | Thyme herb | Azoxystrobin | 70.0 | 0.005 | 0.013 | ±0.003 |
| | | Linuron | 1.0 | 0.005 | 0.021 | ±0.006 |
| | | Metalaxyl | 2.0 * | 0.002 | 0.058 | ±0.017 |
| | | Metalaxyl-M | | 0.002 | 0.060 | ±0.019 |
| 23 | Thyme herb | Azoxystrobin | 70.0 | 0.005 | 0.041 | ±0.008 |
| | | Carbendazim | 0.1 | 0.002 | 0.022 | ±0.007 |
| | | Chlorantraniliprole | 20.0 | 0.005 | 0.094 | ±0.043 |
| | | Chlorotoluron | 0.02 | 0.002 | 0.009 | ±0.002 |
| | | Linuron | 1.0 | 0.005 | 0.062 | ±0.019 |
| | | Metalaxyl | 2.0 * | 0.002 | 0.015 | ±0.004 |
| | | Metalaxyl-M | | 0.002 | 0.013 | ±0.004 |
| | | Metolachlor | 0.05 * | 0.005 | 0.012 | ±0.003 |
| | | Metolachlor S | | 0.002 | <LOQ = 0.002 | ±0.002 |
| 24 | Thyme herb | Azoxystrobin | 70.0 | 0.005 | 0.020 | ±0.004 |
| | | Linuron | 1.0 | 0.005 | 0.016 | ±0.005 |
| 25 | Thyme herb | Azoxystrobin | 70.0 | 0.005 | 0.050 | ±0.01 |
| | | Linuron | 1.0 | 0.005 | 0.100 | ±0.003 |
| 26 | Thyme herb | Azoxystrobin | 70.0 | 0.005 | 0.210 | ±0.040 |
| | | Linuron | 1.0 | 0.005 | 0.026 | ±0.008 |
| 27 | Thyme herb | Linuron | 1.0 | 0.005 | 0.110 | ±0.030 |
| 28 | Thyme herb | Azoxystrobin | 70.0 | 0.005 | 0.009 | ±0.002 |
| | | Linuron | 1.0 | 0.005 | 0.014 | ±0.004 |
| 29 | Thyme herb | Azoxystrobin | 70.0 | 0.005 | 0.290 | ±0.060 |
| | | Linuron | 1.0 | 0.005 | 0.015 | ±0.005 |
| 30 | Thyme herb | Linuron | 1.0 | 0.005 | 0.085 | ±0.026 |
| 31 | Thyme herb | Azoxystrobin | 70.0 | 0.005 | 0.059 | ±0.012 |
| | | Linuron | 1.0 | 0.005 | 0.008 | ±0.002 |
| | | Metalaxyl | 2.0 * | 0.002 | 0.018 | ±0.005 |
| | | Metalaxyl-M | | 0.002 | 0.015 | ±0.005 |
| 32 | Thyme herb | Azoxystrobin | 70.0 | 0.005 | 0.330 | ±0.110 |
| | | Linuron | 1.0 | 0.005 | 0.015 | ±0.005 |
| 33 | Thyme herb | Azoxystrobin | 70.0 | 0.005 | 0.044 | ±0.009 |
| | | Linuron | 1.0 | 0.005 | 0.013 | ±0.004 |
| 34 | Thyme herb | Azoxystrobin | 70.0 | 0.005 | 0.230 | ±0.080 |
| | | Linuron | 1.0 | 0.005 | 0.018 | ±0.006 |
| 35 | Thyme herb | Azoxystrobin | 70.0 | 0.005 | 0.290 | ±0.099 |
| | | Linuron | 1.0 | 0.005 | 0.048 | ±0.015 |
| | | Metalaxyl | 2.0 * | 0.002 | 0.11 | ±0.030 |
| | | Metalaxyl-M | | 0.002 | 0.12 | ±0.040 |
| | | Trifloxystrobin | 15.0 | 0.002 | 0.015 | ±0.003 |

Table 2. *Cont.*

| No. | Food Product | Pesticide Residue | MRL (mg kg⁻¹) | LOQ (mg kg⁻¹) | Concentration (mg kg⁻¹) | Uncertainty (mg kg⁻¹) |
|-----|--------------|-------------------|---------------|----------------|-------------------------|------------------------|
| 36 | Thyme herb | Azoxystrobin | 70.0 | 0.005 | 0.27 | ±0.092 |
|  |  | Linuron | 1.0 | 0.005 | 0.029 | ±0.009 |
| 37 | Blackcurrant leaf | Azoxystrobin | 5.0 | 0.005 | 1.530 | ±0.520 |
|  |  | Linuron | 0.05 | 0.005 | 0.160 | ±0.050 |
|  |  | Tebuconazole | 1.5 | 0.005 | 0.015 | ±0.004 |
| 38 | Blackcurrant leaf | Azoxystrobin | 5.0 | 0.005 | 1.620 | ±0.550 |
|  |  | Clomazone | 0.01 | 0.005 | 0.038 | ±0.010 |
|  |  | Linuron | 0.05 | 0.005 | 0.290 | ±0.090 |
|  |  | Tebuconazole | 1.5 | 0.005 | 0.051 | ±0.013 |
| 39 | Valerian root | Azoxystrobin | 50.0 | 0.005 | 0.210 | ±0.070 |
| 40 | Coriander fruit | Azoxystrobin | 70.0 | 0.005 | 0.009 | ±0.002 |
| 41 | Elderbery flower | Picoxystrobin | 0.01 | 0.005 | 0.009 | ±0.002 |
| 42 | Purple coneflower leaf | Chlorpyrifos | 0.05 | 0.002 | 0.043 | ±0.010 |
| 43 | Sage leaf | Linuron | 1.0 | 0.005 | 0.012 | ±0.004 |
| 44 | Linseed | Epoxiconazole | 0.05 | 0.005 | 0.010 | ±0.003 |
| 45 | Linseed | Chlorpyrifos | 0.05 | 0.005 | 0.050 | ±0.012 |
| Fruits |  |  |  |  |  |  |
| 46 | Blackcurrant | Thiacloprid | 1.0 | 0.002 | 0.060 | ±0.022 |
| 47 | Blackcurrant | Thiacloprid | 1.0 | 0.002 | 0.050 | ±0.019 |
| 48 | Blackcurrant | Fenpyroximate | 1.0 | 0.002 | 0.027 | ±0.009 |
|  |  | Thiacloprid | 1.0 | 0.002 | 0.022 | ±0.004 |
|  |  | Trifloxystrobin | 1.0 | 0.002 | 0.021 | ±0.006 |
| 49 | Blackcurrant | Thiacloprid | 1.0 | 0.002 | 0.016 | ±0.006 |
| 50 | Blackcurrant | Acetamiprid | 2.0 | 0.002 | 0.016 | ±0.003 |
|  |  | Trifloxystrobin | 1.0 | 0.002 | 0.070 | ±0.030 |
| 51 | Blackcurrant | Acetamiprid | 2.0 | 0.002 | 0.023 | ±0.005 |
| 52 | Blackcurrant | Acetamiprid | 2.0 | 0.002 | 0.011 | ±0.002 |
|  |  | Thiacloprid | 1.0 | 0.002 | 0.066 | ±0.024 |
| 53 | Blackcurrant | Fenpyroximate | 1.0 | 0.002 | 0.040 | ±0.013 |
| 54 | Cherry | Dodine | 5.0 | 0.002 | 0.087 | ±0.018 |
|  |  | Thiacloprid | 0.02 | 0.002 | 0.003 | ±0.001 |
| 55 | Cherry | Dodine | 5.0 | 0.002 | 0.037 | ±0.008 |
| 56 | Strawberry | Acetamiprid | 0.5 | 0.001 | 0.005 | ±0.001 |
|  |  | Azoxystrobin | 10.0 | 0.0001 | 0.100 | ±0.034 |
|  |  | Chlorotoluron | 0.01 | 0.0005 | 0.009 | ±0.002 |
|  |  | Cyprodinil | 5.0 | 0.001 | 0.150 | ±0.038 |
|  |  | Difenoconazole | 0.4 | 0.0002 | 0.063 | ±0.016 |
|  |  | Fludioxonil | 4.0 | 0.0001 | 0.200 | ±0.068 |
|  |  | Mepanipyrim | 1.5 | 0.0001 | 0.080 | ±0.026 |
|  |  | Trifloxystrobin | 1.0 | 0.0001 | 0.330 | ±0.092 |
| 57 | Apple | Diflubenzuron | 0.05 | 0.01 | 0.042 | ±0.011 |
|  |  | Fenpyroximate | 0.05 | 0.002 | 0.013 | ±0.004 |
|  |  | Fenpropimorph | 0.05 | 0.002 | 0.011 | ±0.003 |
|  |  | Teflubenzuron | 2.0 | 0.01 | 0.046 | ±0.012 |
|  |  | Triflumuron | 2.0 | 0.002 | 0.063 | ±0.016 |

**Table 2.** *Cont.*

| No. | Food Product | Pesticide Residue | MRL (mg kg⁻¹) | LOQ (mg kg⁻¹) | Concentration (mg kg⁻¹) | Uncertainty (mg kg⁻¹) |
|---|---|---|---|---|---|---|
| 58 | Apple | Acetamiprid | 0.8 | 0.002 | 0.024 | ±0.005 |
| | | Carbendazim | 0.2 | 0.002 | 0.098 | ±0.030 |
| | | Chlorpyrifos | 0.5 | 0.002 | 0.085 | ±0.020 |
| | | Diflubenzuron | 5.0 | 0.01 | 0.014 | ±0.004 |
| | | Fenpyroximate | 0.3 | 0.002 | 0.018 | ±0.006 |
| | | Fludioxonil | 5.0 | 0.002 | 0.013 | ±0.004 |
| | | Methoxyfenozide | 2.0 | 0.002 | 0.064 | ±0.016 |
| | | Pirimicarb | 2.0 | 0.002 | 0.020 | ±0.005 |
| | | Pyraclostrobin | 0.5 | 0.002 | 0.039 | ±0.012 |
| | | Thiacloprid | 0.3 | 0.002 | 0.023 | ±0.009 |
| | | Tebuconazole | 0.3 | 0.005 | 0.025 | ±0.006 |
| | | Trifloxystrobin | 0.5 | 0.002 | 0.025 | ±0.007 |
| 59 | Apple | Carbendazim | 0.2 | 0.002 | 0.062 | ±0.019 |
| 60 | Strawberry | Azoxystrobin | 60.0 | 0.005 | 0.009 | ±0.003 |
| 61 | Strawberry | Fludioxonil | 4.0 | 0.002 | 0.043 | ±0.015 |
| 62 | Raspberry | Imidacloprid | 5.0 | 0.005 | 0.009 | ±0.003 |
| | | Thiamethoxam | 0.05 | 0.002 | 0.009 | ±0.003 |
| 63 | Elderberry | Chlorpyrifos | 0.05 | 0.002 | 0.015 | ±0.003 |
| Spices | | | | | | |
| 64 | Black pepper | Metalaxyl | 0.1 * | 0.002 | 0.016 | ±0.003 |
| | | Metalaxyl-M | | 0.002 | 0.015 | ±0.003 |
| 65 | Black pepper | Acetamiprid | 0.05 | 0.002 | 0.012 | ±0.002 |
| | | Azoxystrobin | 0.3 | 0.005 | 0.022 | ±0.007 |
| | | Carbofuran | 0.05 | 0.002 | 0.015 | ±0.004 |
| | | Metalaxyl | 0.1 * | 0.002 | 0.019 | ±0.004 |
| | | Metalaxyl-M | | 0.002 | 0.018 | ±0.004 |
| 66 | Black pepper | Metalaxyl | 0.1 * | 0.002 | 0.041 | ±0.008 |
| | | Metalaxyl-M | | 0.002 | 0.039 | ±0.008 |
| 67 | Black pepper | Metalaxyl | 0.1 * | 0.002 | 0.011 | ±0.002 |
| | | Metalaxyl-M | | 0.002 | 0.010 | ±0.002 |
| 68 | Orange skin | Imazalil | 5.0 | 0.002 | 3.090 | ±0.772 |
| | | Prochloraz | 10.0 | 0.002 | 0.071 | ±0.018 |
| | | Thiabendazole | 5.0 | 0.005 | 2.020 | ±0.505 |
| 70 | Curcuma | Chlorpyrifos | 1.0 | 0.002 | 0.042 | ±0.010 |
| 69 | Caraway fruit | Chlorpyrifos | 1.0 | 0.002 | 0.051 | ±0.012 |
| 71 | Caraway fruit | Acetamiprid | 0.05 | 0.002 | 0.99 | ±0.200 |
| | | Azoxystrobin | 0.3 | 0.005 | 0.014 | ±0.003 |
| | | Carbendazim | 0.1 | 0.002 | 1.50 | ±0.450 |
| | | Chlorpyrifos | 1.0 | 0.002 | 0.095 | ±0.022 |
| | | Thiamethoxam | 0.05 | 0.002 | 0.100 | ±0.030 |
| 72 | Caraway fruit | Carbendazim | 0.1 | 0.002 | 0.048 | ±0.014 |
| | | Chlorpyrifos | 1.0 | 0.002 | 0.028 | ±0.006 |
| | | Fenpropimorph | 0.1 | 0.002 | 0.011 | ±0.003 |
| Vegetables | | | | | | |
| 73 | Carrot | Chlorpyrifos | 0.1 | 0.002 | 0.013 | ±0.006 |
| 74 | Beetroot | Tebuconazole | 0.02 | 0.005 | 0.008 | ±0.002 |
| 75 | Celery root | Azoxystrobin | 1.0 | 0.005 | 0.007 | ±0.003 |
| | | Linuron | 0.5 | 0.005 | 0.027 | ±0.010 |
| | | Tebuconazole | 0.5 | 0.005 | 0.013 | ±0.004 |
| 76 | Parsley root | Linuron | 0.2 | 0.005 | 0.038 | ±0.013 |
| 77 | Broccoli | Chlorpyrifos | 0.05 | 0.002 | 0.200 | ±0.048 |

**Table 2.** *Cont.*

| No. | Food Product | Pesticide Residue | MRL (mg kg$^{-1}$) | LOQ (mg kg$^{-1}$) | Concentration (mg kg$^{-1}$) | Uncertainty (mg kg$^{-1}$) |
|---|---|---|---|---|---|---|
| 78 | Radish | Metalaxyl | 0.1 * | 0.002 | 0.011 | ±0.004 |
| | | Metalaxyl-M | | 0.002 | 0.011 | ±0.003 |
| | | Pyraclostrobin | 0.5 | 0.002 | 0.016 | ±0.005 |
| 79 | Chive | Azoxystrobin | 70.0 | 0.005 | 0.051 | ±0.024 |
| | | Imidacloprid | 2.0 | 0.005 | 0.009 | ±0.003 |
| | | Linuron | 1.0 | 0.005 | 0.007 | ±0.002 |
| 80 | Dill | Azoxystrobin | 0.3 | 0.005 | 0.028 | ±0.013 |
| | | Chlorpyrifos | 5.0 | 0.002 | 0.019 | ±0.006 |
| | | Mepanipyrim | 0.05 | 0.002 | 0.012 | ±0.004 |
| 81 | Parsley root | Azoxystrobin | 70.0 | 0.005 | 0.050 | ±0.024 |
| | | Chlorpyrifos | 0.05 | 0.002 | 0.220 | ±0.090 |
| **Fruit and vegetable juices** | | | | | | |
| 82 | Pear juice | Acetamiprid | 0.8 | 0.0001 | 0.010 | ±0.004 |
| | | Bosacalid | 2.0 | 0.0005 | 0.015 | ±0.004 |
| | | Clothianidin | 0.4 | 0.0005 | 0.008 | ±0.003 |
| 83 | Beetroot juice | Tebuconazole | 0.02 | 0.0001 | 0.068 | ±0.017 |

LOQ—The Limit of Quantification, MRL—Maximum Residue Levels, * sum of Metalaxyl and Metalaxyl-M.

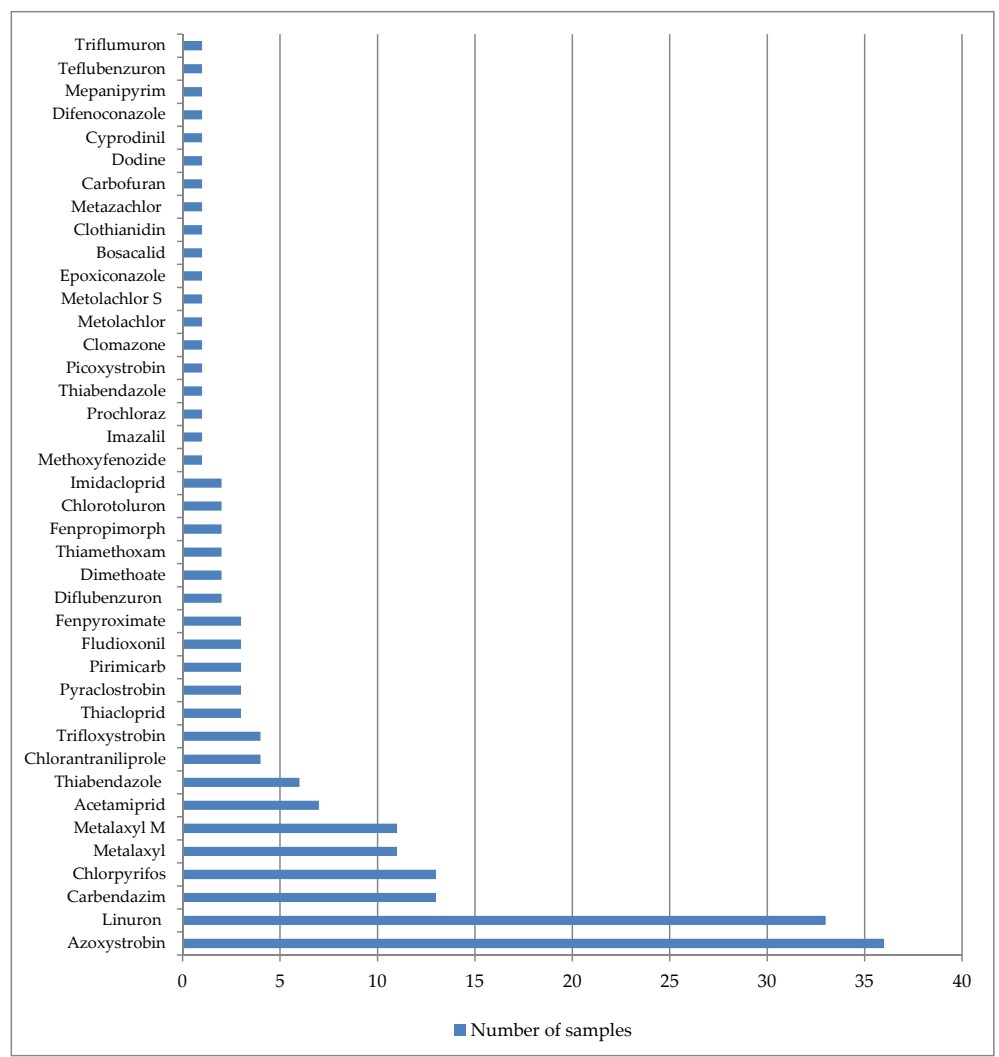

**Figure 1.** Pesticide occurrence frequency in analysed samples.

## 4. Discussion

In the presented study, the percentage share of samples containing pesticide residues (42.9–66.7%) correlates with the results obtained by other authors for the criterion "kind of sample"—Table 3. In studies concerned with vegetables, the percentage share of samples in which pesticide residues were noted varied from 15.9% to 77.8% (Table 1). Similar results were obtained in studies including fruit samples, for which the presence of pesticide residues was from 33.3 % to 77.4% of cases (Table 3). Referring to earlier results from studies covering samples of fruits and vegetables, it was demonstrated that pesticide residues in vegetables were less frequently found than in fruits [13,14], which is also supported by the results obtained in the presented experiment. Similar data were published by the European Food Safety Authority (EFSA) in 2014 and 2015, in the area of control studies on pesticide residues in food products in the member states of the European Union, indicating the presence of pesticide residues in 49–53% of samples of vegetables. Comparative studies on conventional and organic cultivations also confirmed a higher frequency of occurrence of pesticide residues in samples of fruits (75% and 25.8%) in relation to samples of vegetables (32% and 8.7%) [15,16]. The cause for this is attributed by those authors to the probability of application of a higher concentration of plant protection agents with extended effect duration, as well as to the use of various spraying technologies which may contribute to an increased accumulation of pesticide residues in fruits. A compilation of numerical data concerning the observed presence of various pesticide residues in food samples is presented in Figure 2. In the study, the own group of pesticides was most often determined as fungicides—47.5%, while every third designated plant protection product was an insecticide (32.5%). Fungicides dominated in samples from domestic primary production, tested by Dyjak et al. [17] in 2017 and Nowacka et al. [18] in 2011, as they constituted 45.5% and 63.9% respectively, and insecticides—24.5% and 32.5%. Also, in studies conducted by Szpyrk et al. [19], fungicides occurred as the most common pesticide residues. Analysing the frequency of occurrence of various pesticide residues in samples of fruits and vegetables (Figure 2), the most frequently identified pesticides were: chlorpyrifos (25%), cypermethrin (16.7%), imazalil (16.7%), azoxystrobin (12.5%), carbendazim (12.5%), imidacloprid (8.3%), cyprodinil (8.3%), permethrin (8.3%) and pyridaben (8.3%), enosulfan (4.2%), difenoconazole (4.2%), haloxyfop-R-Methyl (4.2%), boscalid (4.2%), chlorothalonil (4.2%), phosalone (4.2%), $\sum$-HCH (4.2%), diazinon (4.2%), enthoprophos (4.2%), pendimethalin (4.2%), acequinocyl (4.2%), iprodione (4.2%), bifenthrin (4.2%), deltamethrin (4.2%), metalaxyl (4.2%), and thiabendazole (4.2%). Four of those—azoxystrobin, carbendazim, chlorpyrifos, and metalaxyl—were also among the most frequently identified pesticides in the presented study (Figure 1). Authors conducting research on the presence of pesticide residues in plant samples also confirm the presence of those pesticides in samples of fruits and vegetables, in food of plant origin, in diet supplements, and also in plant samples used in Chinese medicine (Table 3). In the group of analysed fruits, pesticide residues were most frequently identified in samples of blackcurrant (44.4%), which is also reported in a study conducted in Poland in the years 2010–2015, in which the highest percentage level of pesticide residues among all of the analysed samples was demonstrated in blackcurrant—50% [15,20], and in black and red currant—40.9% [14]. In the presented study, the level of detected pesticide residues in herbs (52.9%) and spices (42.7%) correlates with the results obtained in study by Reinholds et al. [21] and Kowalska [11], who demonstrated the presence of pesticides in 59% and 71% of analysed samples of herbs and spices. The number of detected pesticide residues in herbs varied from 1 to 7 compounds in an individual sample of thyme (Table 2). Only in 3 (7.7%) among the 39 analysed samples of thyme was no presence found of the plant protection agents from the group analysed in their experiment. The remaining 36 samples contained pesticides, which is also confirmed by the study of Reinholds et al. [21], which showed similar values of pesticide residues in the analysed samples of that raw material (82%). In our study, the most frequently identified pesticides were linuron and azoxystrobin, while in studies by other authors, presence of plant procection, such as cymoxanil, dimethoate, and tebuconazole, was found [21–23]. Studies conducted in Poland have demonstrated the presence of the same pesticides in the analysed herb samples—azoxystrobin and linuron [11].

In the group of analysed spices, the sought compounds were detected most frequently in samples of black pepper (44.4%). In 4 out 7 analysed samples, the presence of pesticide residues was found, which is supported by the study by Ferrer-Amate et al. [24], and Reinholds et al. [21], who obtained similar results for samples of that spice.

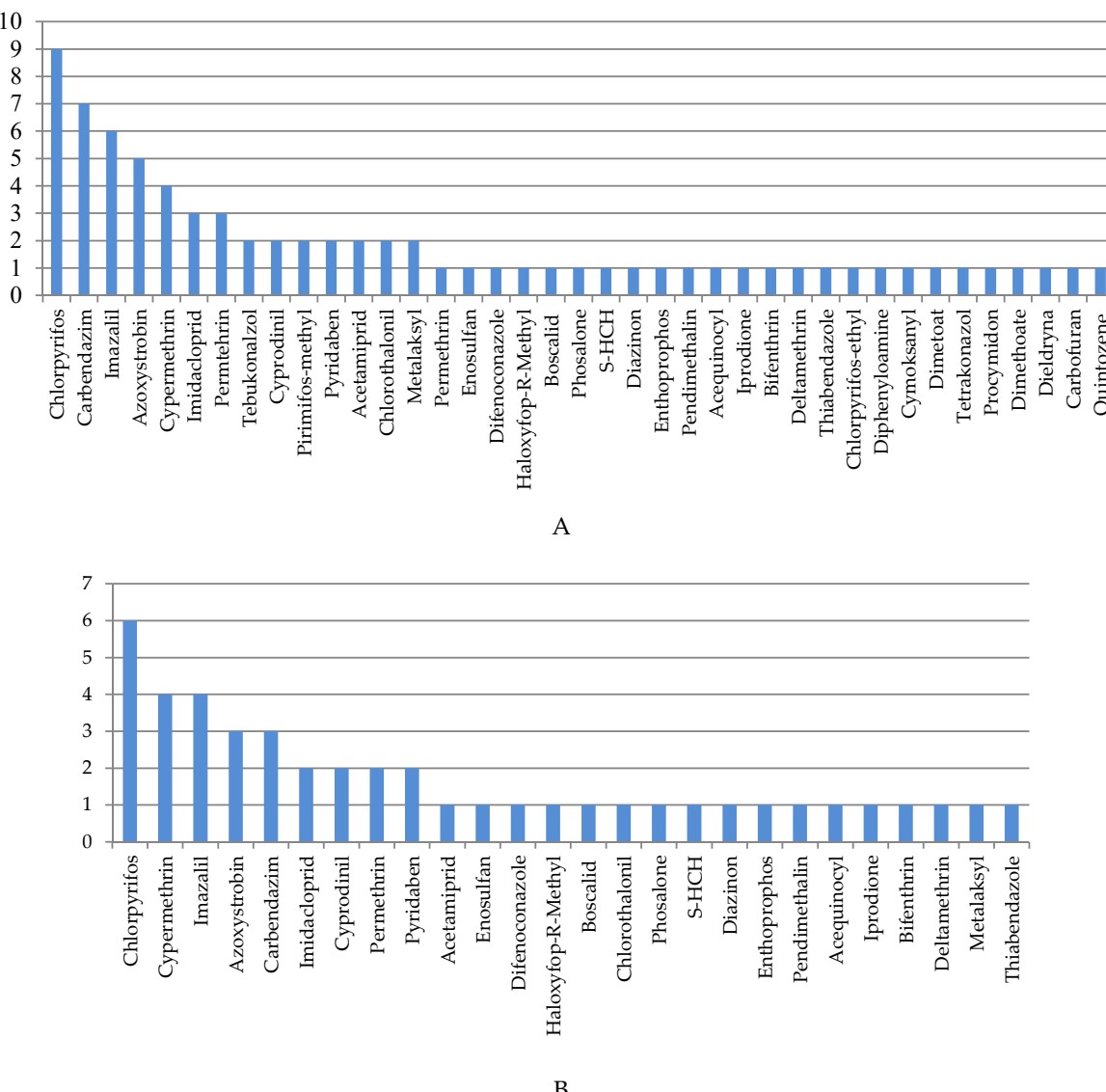

**Figure 2.** The most frequently detected pesticide residues in samples of plant origin according to literature data, (**A**) in all samples, (**B**) in fruit and vegetable samples (literature reports in Table 3).

**Table 3.** Summary of the most frequently detected pesticides in different food samples reported in the literature.

| No. | Food Category | No. of Samples | No. of Samples with Detected Residues | No. of Analysed Pesticides | No. of Detected Pesticides | Most Frequently Found Pesticide | % [1] | % [2] | References |
|---|---|---|---|---|---|---|---|---|---|
| 1 | Vegetables | 1057 | 168 | 86 | 43 | Permethrin Enosulfan | 15.9 | 50.0 | [25] |
| 2 | Vegetables | 30 | 5 | 283 | 4 | Cypermethrin Chlorpyrifos Difenoconazole | 16.7 | 1.4 | [26] |
| 3 | Vegetables (bean) | 178 | 39 | 58 | 39 | Cyprodinil, Haloxyfop-R-Methyl | 21.9 | 67.2 | [27] |
| 4 | Vegetables | 365 | 118 | 130 | 15 | Chlorpyrifos Cypermethrin | 32.3 | 11.5 | [28] |
| 5 | Vegetables | 138 | 47 | 242 | 17 | Azoxystrobin Boscalid Chlorothalonil | 34.1 | 7.0 | [29] |
| 6 | Vegetables (tomato) | 20 | 8 | 30 | 6 | Azoxystrobin Cyprodinil | 40.0 | 20.0 | [8] |
| 7 | Vegetables | 90 | 70 | 18 | 14 | Chlorpyrifos Phosalone | 77.8 | 77.8 | [10] |
| 8 | Vegetables | 20 | Not defined | 48 | 23 | Σ-HCH, Permethrin | - | 47.9 | [30] |
| 9 | Fruit (peach) | 1150 | 383 | 31 | 22 | Chlorpyrifos Diazinon | 33.3 | 71.0 | [31] |
| 10 | Omija fruit and juice | 420 | 143 | 33 | 4 | Enthoprophos Pendimethalin | 34.1 | 12.1 | [32] |
| 11 | Yuza fruits and tea | 155 | 120 | 7 | 3 | Carbendazim Acequinocyl | 77.4 | 42.9 | [7] |
| 12 | Fruits and vegetables | 199 | 46 | 74 | Not defined | Imazalil Iprodione Azoxystrobin | 23.1 | - | [33] |
| 13 | Fruits and vegetables | 20 | 5 | 82 | 36 | Pyridaben | 25.0 | 43.9 | [34] |
| 14 | Fruits and vegetables | 144 | 46 | 60 | 15 | Carbendazim Acetamiprid | 31.9 | 25.0 | [20] |
| 15 | Fruits and vegetables | 866 | 293 | 102 | 30 | Imazalil | 33.8 | 29.4 | [35] |

**Table 3.** *Cont.*

| No. | Food Category | No. of Samples | No. of Samples with Detected Residues | No. of Analysed Pesticides | No. of Detected Pesticides | Most Frequently Found Pesticide | % [(1)] | % [(2)] | References |
|-----|---------------|----------------|----------------------------------------|-----------------------------|-----------------------------|----------------------------------|---------|---------|------------|
| 16 | Fruits and vegetables | 3009 | 1135 | 22 | 22 | Cypermethrin | 37.7 | 100.0 | [36] |
| 17 | Fruits and vegetables | 1463 | 689 | 121 | 44 | Bifenthrin Pyridaben | 47.1 | 36.4 | [37] |
| 18 | Fruits and vegetables | 13,556 | 6548 | 229 | 15 | Carbendazim Chlorpyrifos | 48.3 | 6.6 | [6] |
| 19 | Fruits and vegetables | 150 | 88 | 34 | 16 | Deltamethrin Imidacloprid Cypermethrin Chlorpyrifos Metalaksyl | 58.7 | 47.1 | [38] |
| 20 | Fruits and vegetables | 724 | 586 | 326 | 83 | Thiabendazole Imazalil | 80.9 | 25.5 | [5] |
| 21 | Fruits and vegetables | 17 | 17 | 100 | 26 | Imazalil Imidacloprid | 100.0 | 26.0 | [9] |
| 22 | Herbs | 30 | 2/3 | 155 | 3 | Chlorpyrifos-ethyl Diphenyloamine Tebukonazol | 6.7–10.0 | 1.9 | [22] |
| 23 | Herbs and spices | 300 | 177 | 134 | 24 | Cymoksanyl Dimetoat Tebukonazol Tetrakonazol | 59.0 | 17.9 | [21] |
| 24 | Herbs | 104 | 75 | 250 | 16 | Azoxystrobin Linuron Carbendazim | 72.1 | 6.4 | [11] |
| 25 | Foods of plant origin and drinks | 126 | 42 | 47 | 18 | Chlorpyrifos Procymidon Primifos-methyl Dimethoate Dieldryna | 33.3 | 38.3 | [39] |

**Table 3.** *Cont.*

| No. | Food Category | No. of Samples | No. of Samples with Detected Residues | No. of Analysed Pesticides | No. of Detected Pesticides | Most Frequently Found Pesticide | % [1] | % [2] | References |
|---|---|---|---|---|---|---|---|---|---|
| 26 | Fruit juices | 106 | 46 | 53 | 9 | Carbendazim Imazalil | 43.4 | 17.0 | [3] |
| 27 | Fruit juices | 21 | 10 | 174 | 21 | Imidacloprid Acetamiprid | 47.6 | 12.0 | [40] |
| 28 | Fruit-based soft drinks | 94 | 85 | 30 | 11 | Carbendazim Imazalil | 90.4 | 36.7 | [4] |
| 29 | Cereals | 89 | 14 | 110 | 3 | Primifos-methyl | 15.7 | 2.7 | [41] |
| 30 | Cereals | 380 | 145 | 292 | Not defined | Permethrin Tebukonazol | 38.0 | - | [23] |
| 31 | Chinese herbal medicines | 294 | 108 | 162 | 42 | Chlorpyrifos | 36.7 | 25.9 | [42] |
| 32 | Plant used in traditional Chinese medicine | 138 | 95 | 116 | 55 | Carbendazim Carbofuran | 68.8 | 47.4 | [43] |
| 33 | Traditional Chinese medicine | 20 | 20 | 55 | 6 | Quintozene Chlorothalonil Chlorpyrifos | 100.0 | 10.9 | [44] |
| 34 | Dried botanical dietary supplements | Not defined | Not defined | 236 | 73 | Carbendazim Metalaxyl Azoxystrobin | - | 30.9 | [45] |
| 35 | Food samples | 31 | 9 | 44 | 8 | Acetamiprid Azoxystrobin | 29.0 | 18.8 | [46] |

[1] The percentage of total number of analysed sample to the total number of detected pesticides. [2] The percentage of detected pesticides to the total number of pesticides analysed.

The literature review revealed the presence of metalaxyl and carbendazim in samples of black pepper, which was also observed in our experiment. In none of the analysed samples of herbs were exceeded levels of concentration (above the MRL) observed, which does not support the results obtained by Reinholds et al. [21] and Kowalska [11], where the concentrations of pesticide residues in 10% [21] of samples of oregano and in 46% [21] and 15% [11] of samples of thyme were above the permissible values. The literature review, in the aspect of the content of pesticide residues in samples of juices, demonstrated that the percentage share of samples in which the sought compounds were identified varied from 43.40% to 90.43% [3,4,40], which is in conformance with the results obtained in this study for the samples of fruit and vegetable juices—50%. In our own study, the most frequently assayed pesticides were acetamiprid, boscalid, clothianidin, and tebucanozole (Table 2), while in the literature reports—acetamiprid, carbendazim, and imazalil (Table 3). In the analysed samples of cereals, no presence of pesticide residues was found. Literature data concerning studies on pesticide residues in cereals in Poland in the years 2009–2013 report the presence of those compounds in the range from 15.73% to 38% of the analysed samples [23,41]. In the presented study, only 3 cereal samples were analysed, which constituted as little as 1.9% of the total number of analysed samples, and that number did not constitute a representative value in relation to the remaining kinds of samples. Summing up the results obtained in this study, it should be emphasised that 51.9% of the samples of plant materials and food products originating from the eastern part of Poland contained pesticide residues, but their levels did not exceed the higher permissible concentrations. Most frequently, pesticide residues were detected in fruit samples (66.7%), compared to the remaining groups of analysed products, where the percentage share of samples containing the sought compounds was at the level of approximately 50% in each group. Special note should be taken of the possible contamination with thiacloprid and trifloxystrobin in fruits of blackcurrant, carbendazim in apples, and azoxystrobin and fludioxonil in strawberries. The analysed samples of fruits contained the largest number and diversity of identified pesticide residues, compared to the remaining samples, which raises concern relating to the quality of those food components. Pesticide cocktails found in food pose a serious threat to people and the environment. Mixtures of pesticides can have far more harmful effects than exposure to individual chemicals, both in humans and other species, such as insects, fish, and birds [47,48]. Pesticides are found in millions of different combinations at different concentrations in our food and landscape. It is probably impossible to create a system sufficiently advanced to be able to assess the full spectrum of health and environmental effects resulting from long-term exposure to hundreds of different pesticides. The results of this study emphasise the importance of monitoring of pesticide residues in herbs and spices, especially in the case of thyme and black pepper, which were identified as the most contaminated matrices in that group of products, in which the percentage share of samples containing pesticide residues was at the level of 80% and 44%, respectively.

## 5. Conclusions

Studies in the area of analysis of pesticide residues are highly important in the estimation of quality of raw materials of plant origin, as well as food. The results obtained in this study indicate that the occurrence of pesticide residues in the analysed products cannot be considered to be a serious threat to human and animal health. Nevertheless, constant monitoring of the content of pesticide residues and strict regulations concerning the highest permissible concentrations of those compounds in food samples are of key importance for the alleviation of potential risk to the health and life of consumers. Due to the harmful effects of the cocktail effect of pesticides, perhaps the only way to minimise the risks to health and the environment is to significantly reduce the overall use of pesticides. There is also a need to introduce urgently needed measures to support farmers to significantly reduce pesticide use and switch to organic farming systems in which synthetic pesticides are replaced by botanical pesticides or chemical control is completely avoided.

**Supplementary Materials:** The following are available online at http://www.mdpi.com/2077-0472/10/6/192/s1, Table S1: List of pesticides determined in the positive ionization mode, Table S2: List of pesticides determined in the negative ionization mode.

**Author Contributions:** G.K. and R.K. conceived the research idea and experimental protocol; G.K. coordinated the research; G.K. and R.K. wrote the manuscript; G.K. and R.K. managed writing—review and editing; G.K., U.P., and R.K. had the supervision task; G.K., U.P., and R.K. were involved in crop management and performed the determinations of biochemical and physiological analyses; G.K. managed the data statistical processing; G.K., U.P., and R.K. were involved in bibliographic search. All authors have read and agreed to the published version of the manuscript.

**Funding:** This work was financed by a statutory activity subsidy from the Polish Ministry of Science and Higher Education for the Faculty of Agrobioengineering of University of Life Sciences in Lublin and for the Faculty of Food Science and Biotechnology of University of Life Sciences in Lublin.

**Conflicts of Interest:** The authors declare that they have no conflicts of interest to disclose.

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
