# Peer review of "Estimation of Pesticide Residues in Selected Products of Plant Origin from Poland with the Use of the HPLC-MS/MS Technique"

_agriculture, doi:10.3390/agriculture10060192_

Round 1
Reviewer 1 Report
Major concerns
1) If the procedure is approved by the Polish Centre of Accreditation, the validation of the method should be revealed to the readers.
- I doubt that used gradient elution offers acceptable retention time stability. Mainly due to the fact that after return of organic solvent from 100% to 20%, analytical column is equilibrated to initial conditions in 3 min! Seems to be too short.
- I doubt that quantitative analses of 250 substances can be based on the use of two internal standards, only. How is it possible to achieve acceptable precision of the analytical method with 2 internal standards that could hardly represent physicochemical properties and consequently mass spectrometric behavoir of all 250 analytes?
2) Clerify how you have selected 250 pesticides and why.
3) I miss clear interpretation of the results. Data obtained should be used to uncover not only farmes' attitude towards consumers, but also attitude towards nature and environment. In this respect, use obtained data to describe whether it can be concluded that producers follow the rules of utilization of registered pesticides in order to prevent resistance as well as the rules for utilization of particular pesticide for specific crops, only. How often are these rules overcome?
4) Please, discuss also to what extent the cocktail effect is a problem and the potential impact upon human health and the environment.
Minor concerns
1) Use appropriate unit for centrifugation (rpm are not acceptable, since you do not state radius of rotor).
2) eluant vs eluent
3) Make sentence line 260-262 readable. It is not clear who performed the experiment - you or Reinholds et al.
4) Please, explain what it means that ..."In our study the most..., while in the studie of other authors....tebuconazole." (line 265-266).
5) make number of digits in case of % values uniform throughout the manuscript
6) correct the sentence line 283-286
Author Response
Dear Reviewer
We would like to thank you for thoroughly reviewing the manuscript and their thoughtful criticism. We appreciate your suggestions and have revised the manuscript to address the comments and followed all suggestions to strengthen the manuscript.
We hope that our corrections will prove suitable enough for the demands of the publisher. We are still open to all further critical remarks and suggestions.
We look forward to hearing from you.
Major concerns
- If the procedure is approved by the Polish Centre of Accreditation, the validation of the method should be revealed to the readers.
- I doubt that used gradient elution offers acceptable retention time stability. Mainly due to the fact that after return of organic solvent from 100% to 20%, analytical column is equilibrated to initial conditions in 3 min! Seems to be too short.
- I doubt that quantitative analses of 250 substances can be based on the use of two internal standards, only. How is it possible to achieve acceptable precision of the analytical method with 2 internal standards that could hardly represent physicochemical properties and consequently mass spectrometric behavoir of all 250 analytes? – The analyzes were carried out in an accredited laboratory, so the method and results meet international quality requirements (accreditation certificate AB 1375 https://www.pca.gov.pl/en/accredited-organizations/accredited-organizations/testing-laboratories/AB%201375,entity.html). The laboratory performing pesticide determinations has proven competence. Please find attached the analytical scope (pp. 6-10). Due to the requirements of the certification authority, all analyzed pesticides were characterized by recovery in the range of 70% to 120%. The limiting criterion for linearity was the range above r³995 (values from 0.9950 to 0.9998 were obtained). The method used is in accordance with PN-EN 15662: 2008, so we did not introduce changes to this standard. Two internal standards were applied in accordance with PN-EN 15662: 2008 (Annex) and the correct values of validation parameters were obtained, which confirms the accreditation. Method optimization is not the topic of the article. The more that the method is standardized. Calibration curves were made in a clean matrix for tested matrices, which were previously prepared by the quechers method. Calibration was performed using internal standards (IS) - TPP in positive ionization mode and BNPU in negative ionization mode. To prepare the pesticide solution (Master Mix), stock solutions were made containing 15-20 compounds in a concentration of 1 mg / ml or 10 µg / ml. The Master Mix was made by mixing the stock solutions and supplementing them with a pure matrix extract, respectively, to obtain a concentration of 100 ng / ml.
- Clerify how you have selected 250 pesticides and why. –
- First of all, the choice of analyzed pesticides resulted from the demand of herbs producers' customers for analyzes in line with the laboratory services market in the region. In addition, only pesticides for which the criteria for analytical quality were met were included in the analysis.
- I miss clear interpretation of the results. Data obtained should be used to uncover not only farmes' attitude towards consumers, but also attitude towards nature and environment. In this respect, use obtained data to describe whether it can be concluded that producers follow the rules of utilization of registered pesticides in order to prevent resistance as well as the rules for utilization of particular pesticide for specific crops, only. How often are these rules overcome?–
- In the presented research, no sample of plant origin did not contain residues of a pesticide that is not authorized for trading in Poland or for the protection of a given plant species.
- 4. Please, discuss also to what extent the cocktail effect is a problem and the potential impact upon human health and the environment.
- Pesticide cocktails found in food pose a serious threat to people and the environment. Mixtures of pesticides can have far more harmful effects than exposure to individual chemicals, both in humans and other species such as insects, fish and birds. [43,44]. Pesticides are found in millions of different combinations at different concentrations in our food and landscape. It is probably impossible to create a system sufficiently advanced to be able to assess the full spectrum of health and environmental effects resulting from long-term exposure to hundreds of different pesticides.
Due to the harmful effects of the cocktail effect of pesticides, perhaps the only way to minimize the risks to health and the environment is to significantly reduce the overall use of pesticides. There is also a need to introduce urgently needed measures to support farmers in order to significantly reduce the use of pesticides and switch to agro-organic farming systems using Integrated Pest Management (IPM).
Minor concerns
- 1. Use appropriate unit for centrifugation (rpm are not acceptable, since you do not state radius of rotor) - 3000 rpm, the rotor radius is 135 mm, 1361 rcf
- 2. eluant vs eluent – corrected as recommended by the reviewer.
- Make sentence line 260-262 readable. It is not clear who performed the experiment - you or Reinholds et al – corrected as recommended by the reviewer.
- Please, explain what it means that ..."In our study the most..., while in the studie of other authors....tebuconazole." (line 265-266) – corrected as recommended by the reviewer.
- make number of digits in case of % values uniform throughout the manuscript – corrected as recommended by the reviewer.
- correct the sentence line 283-286 - corrected as recommended by the reviewer.
- The English is readable, but some sentences are not finished and in some cases grammar and spelling errors occurs. The help of native English speaker would help to overcome this issue – language correction of text (grammar, usage, and overall readability of the manuscript) I have entrusted my colleague PaweÅ‚ Pasikowski (Translator and Interpreter, English Language Instructor).
Yours sincerely,
Radosław Kowalski

Reviewer 2 Report
Kowalska. et al. analyzed the content of 250 pesticides in 160 samples of plant origin. The authors utilized a standard QuECheRS extraction protocol followed by analysis on an LC-MS/MS system. The work is interesting, and overall, the manuscript is prepared and written adequately. The introduction is rather short and general. Perhaps, it would benefit from the addition of references focusing on the same country/ region to strengthen the rationale behind this study. The description in the methodology section is sufficient. Results, in certain cases, can be presented more clearly. The discussion section, especially table 5, is well prepared. Nevertheless, some points need to be addressed to improve this manuscript further.
The authors should cite their just accepted manuscript on pesticide residues in herbs. Moreover, a short text clarifying and or comparing the difference between the samples/results presented in this study and the published study must be introduced in the manuscript.
Also, as the methodology is practically common for both manuscripts, care must be taken to avoid self-plagiarism issues.
The authors should provide a rationale behind the selection of these pesticides and samples. Regarding the samples, it appears that the number of herb samples and especially thyme (39) and linseed (17) are disproportional to the rest. Is there a specific reason behind this selection?
Line 164: how did the authors evaluate the recovery rate?
Tables 1 and 2: Perhaps it would be better if these tables are provided as supplementary data. Their length could be shortened, as the two different transition ions (and their associated collision energies) could be presented in one line for each pesticide.
Figure 1 and Table 3 are based on the same data: Perhaps, they could be summarized in one table with the addition of the % occurrence.
Figure 1 and Table 3: Although the total number of cereal samples was rather small, can the authors' comment/explain the absence of residues in these samples?
Table 4: Data presented in this table may be rather unclear to the reader. Since the MRL and LOQs for individual pesticides are common, perhaps it would be better if these values are not repeated for each line. Similarly, it can be rather unclear to the reader what is the difference between the 38 thyme samples presented. Perhaps, thyme and other multiple samples can be summarized in one line. Instead of the current form, a column with the range (min-max) and the number of detected samples for each pesticide could be introduced.
Values can be presented as μg/Kg with fewer decimal places. Concentration and uncertainty can be presented in one column.
How were the LOQs defined and what was the total number of samples evaluated for the uncertainty?
Figures 2 and 3. The legends of these figures can be adjusted so that it is clear to the reader that they are presenting data from this and previous literature, respectively.
The English language requires, in certain cases, some improvement. Most of the text is fine, but there are sentences which may appear as incomplete or unclear. Some minor mistakes within the text: i.e line 147 nitro en, line 149 introduce the degree symbol for temperature should also be carefully reviewed.
Author Response
Dear Reviewer
We would like to thank you for thoroughly reviewing the manuscript and their thoughtful criticism. We appreciate your suggestions and have revised the manuscript to address the comments and followed all suggestions to strengthen the manuscript.
We hope that our corrections will prove suitable enough for the demands of the publisher. We are still open to all further critical remarks and suggestions.
We look forward to hearing from you.
Comments:
Kowalska. et al. analyzed the content of 250 pesticides in 160 samples of plant origin. The authors utilized a standard QuECheRS extraction protocol followed by analysis on an LC-MS/MS system. The work is interesting, and overall, the manuscript is prepared and written adequately. The introduction is rather short and general. Perhaps, it would benefit from the addition of references focusing on the same country/ region to strengthen the rationale behind this study. The description in the methodology section is sufficient. Results, in certain cases, can be presented more clearly. The discussion section, especially table 5, is well prepared. Nevertheless, some points need to be addressed to improve this manuscript further.
General comments:
- The authors should cite their just accepted manuscript on pesticide residues in herbs. Moreover, a short text clarifying and or comparing the difference between the samples/results presented in this study and the published study must be introduced in the manuscript – corrected as recommended by the reviewer.
- Also, as the methodology is practically common for both manuscripts, care must be taken to avoid self-plagiarism issues. – the methodology provides a reference to a previously published work
- The authors should provide a rationale behind the selection of these pesticides and samples. Regarding the samples, it appears that the number of herb samples and especially thyme (39) and linseed (17) are disproportional to the rest. Is there a specific reason behind this selection?
- First of all, the choice of analyzed pesticides resulted from the demand of products of plant origin producers' customers for analyzes in line with the laboratory services market in the region. In addition, only pesticides for which the criteria for analytical quality were met were included in the analysis.
- Line 164: how did the authors evaluate the recovery rate?
The method for determining the content of pesticides is an accredited method (accreditation certificate AB 1375 https://www.pca.gov.pl/en/accredited-organizations/accredited-organizations/testing-laboratories/AB%201375,entity.html). The laboratory performing pesticide determinations has proven competence. Please find attached the analytical scope (pp. 6-10). The method is certified. For herbs, fruits, vegetables, fruit juices matrices were tested. Due to the requirements of the certification authority, all analyzed pesticides were characterized by recovery in the range of 70% to 120%. The limiting criterion for linearity was the range above r³0.995 (values from 0.9950 to 0.9998 were obtained). Calibration curves were made in a clean matrix for tested matrices, which were previously prepared by the quechers method. Calibration was performed using internal standards (IS) - TPP in positive ionization mode and BNPU in negative ionization mode. To prepare the pesticide solution (Master Mix), stock solutions were made containing 15-20 compounds in a concentration of 1 mg / ml or 10 µg / ml. The Master Mix was made by mixing the stock solutions and supplementing them with a pure matrix extract, respectively, to obtain a concentration of 100 ng / ml.
- Tables 1 and 2: Perhaps it would be better if these tables are provided as supplementary data. Their length could be shortened, as the two different transition ions (and their associated collision energies) could be presented in one line for each pesticide - corrected as recommended by the reviewer - moved to supplementary data.
- Figure 1 and Table 3 are based on the same data: Perhaps, they could be summarized in one table with the addition of the % occurrence – corrected as recommended by the reviewer.
- Table 4: Data presented in this table may be rather unclear to the reader. Since the MRL and LOQs for individual pesticides are common, perhaps it would be better if these values are not repeated for each line. Similarly, it can be rather unclear to the reader what is the difference between the 38 thyme samples presented. Perhaps, thyme and other multiple samples can be summarized in one line. Instead of the current form, a column with the range (min-max) and the number of detected samples for each pesticide could be introduced. Values can be presented as μg/Kg with fewer decimal places. Concentration and uncertainty can be presented in one column - Thank you for your attention. However, after analyzing Table 4, we decided to leave the current version. Detailed data allows for better interpretation of results.
- How were the LOQs defined and what was the total number of samples evaluated for the uncertainty? The method used is in accordance with PN-EN 15662: 2008, so we did not introduce changes to this standard. Method optimization is not the topic of the article. The more that the method is standardized. “LOD - The smallest point of the calibration curve for which the S / N (signal to noise) ratio is greater than or equal to 5 is the detection limit for the test compound. The S / N ratio is determined using the Analyst software using the Savitzky-Golay Smooth and Signal-to-Noise scripts”. “LOQ - The smallest point of the calibration curve for which the S / N (signal to noise) ratio is greater than or equal to 10 is the limit of quantification for the test compound. The S / N ratio is determined analogously to LOD using the Analyst software”.
- Figures 2 and 3. The legends of these figures can be adjusted so that it is clear to the reader that they are presenting data from this and previous literature, respectively - – corrected as recommended by the reviewer.
- The English language requires, in certain cases, some improvement. Most of the text is fine, but there are sentences which may appear as incomplete or unclear. – language correction of text (grammar, usage, and overall readability of the manuscript) I have entrusted my colleague PaweÅ‚ Pasikowski (Translator and Interpreter, English Language Instructor).
- Some minor mistakes within the text: i.e line 147 nitro en, line 149 introduce the degree symbol for temperature should also be carefully reviewed. – corrected as recommended by the reviewer.
Yours sincerely,
Radosław Kowalski

Round 2
Reviewer 1 Report
Ad 1) I recommend, the validation of the method should be revealed to the readers at least in supplementary material. I must insist that eqilibration time is too short (1,5 of column volume). It seems to be insufficient to equilibrate analytical column to initial conditions. Consequently retention time stability might be compromised. Please, show retention time stability of several consecutive injection of the firts five analytes. It would be relevant scientific argument instead of reffering to accreditation certificate. I agree that method validation is not the topic of the article. However, relevance of your data are tightly linked to the method used. You should either reffer to the published (peer-reviewed method) or show its reliability (via validation).
Ad2) I accepted to review the article on behalf of potential readers. Please, add the explanation to the main text. It is important to reveal the experimental design of the research to the readers not just to satisfy the referee...
Ad3) This is the weakest part of the paper. It is very descriptive, but there is no real discussion. It should not be about a comparison with previously published data only, but also about the interpretation. What are actual meanings of the values presented. No interpretation is available throughout the text. It is pity. Paper is nicely written and data obtained hide much more information than authors present. Analysed pesticides where not even grouped for detail description of their application. The way of data presentation and interpretation are not reflecting agricultural community requirements. What data say about agricultural practice?
Correct the sentence line 205-206 - "In the presented research, no sample of plant origin did not contain residues of a pesticide that is not authorized for trading in Poland or for the protection of a given plant species."
Ad4) OK. The only weak point is the Integrated Pest Management. The control methods in IPM are - cultural, physical, biological and chemical. If you used IPM in context of organic farming, it should be stressed that in such case synthetic pesticides are replaced by botanical pesticides or that chemical control is avoided entirely (line 314-315).
Minor concerns:
Ad 1) rcf (or even better x g) does the job. In such situation info on rpm and rotor radius are redundant.
correct typo line 273
Author Response
Dear Reviewer
We would like to thank you for thoroughly reviewing the manuscript and their thoughtful criticism. We appreciate your suggestions and have revised the manuscript to address the comments and followed all suggestions to strengthen the manuscript.
We hope that our corrections will prove suitable enough for the demands of the publisher. We are still open to all further critical remarks and suggestions.
We look forward to hearing from you.
Ad 1) I recommend, the validation of the method should be revealed to the readers at least in supplementary material. I must insist that eqilibration time is too short (1,5 of column volume). It seems to be insufficient to equilibrate analytical column to initial conditions. Consequently retention time stability might be compromised. Please, show retention time stability of several consecutive injection of the firts five analytes. It would be relevant scientific argument instead of reffering to accreditation certificate. I agree that method validation is not the topic of the article. However, relevance of your data are tightly linked to the method used. You should either reffer to the published (peer-reviewed method) or show its reliability (via validation).
As the reviewer pointed out in the last sentence: at the beginning of the methodology we state that the analysis of pesticides is consistent with the previously described (reviewed) methodology. In addition, I enclose a list of retention times for subsequent injections of 5 pesticides, where it can be seen that the retention times are stable.
Ad2) I accepted to review the article on behalf of potential readers. Please, add the explanation to the main text. It is important to reveal the experimental design of the research to the readers not just to satisfy the referee... – corrected as recommended by the reviewer.
Ad3) This is the weakest part of the paper. It is very descriptive, but there is no real discussion. It should not be about a comparison with previously published data only, but also about the interpretation. What are actual meanings of the values presented. No interpretation is available throughout the text. It is pity. Paper is nicely written and data obtained hide much more information than authors present. Analysed pesticides where not even grouped for detail description of their application. The way of data presentation and interpretation are not reflecting agricultural community requirements. What data say about agricultural practice? – corrected as recommended by the reviewer.
In terms of the use of the marked substances, they were classified into groups: fungicides (47.5%), insecticides (32.5%), herbicides (15%), carbamates (2.5%) and organophosphorus pesticides (2.5%).
In the study, the own group of pesticides was most often determined fungicides - 47.5%, while every third designated plant protection product was an insecticide (32.5%). Fungicides dominated in samples from domestic primary production, tested by Dyjak et al. [17] in 2017 and Nowacka et al. [18] in 2011, as they constituted 45.5% and 63.9% respectively, and insecticides - 24.5% and 32.5%. Also in studies conducted by Szpyrk et al. [19] fungicides occurred as the most common pesticide residues.
Correct the sentence line 205-206 - "In the presented research, no sample of plant origin did not contain residues of a pesticide that is not authorized for trading in Poland or for the protection of a given plant species." – corrected as recommended by the reviewer.
In the presented research, all identified pesticide residues are authorized in Poland. All pesticides found in individual products of plant origin are dedicated to the protection of a given plant species.
Ad4) OK. The only weak point is the Integrated Pest Management. The control methods in IPM are - cultural, physical, biological and chemical. If you used IPM in context of organic farming, it should be stressed that in such case synthetic pesticides are replaced by botanical pesticides or that chemical control is avoided entirely (line 314-315). – corrected as recommended by the reviewer.
There is also a need to introduce urgently needed measures to support farmers to significantly reduce pesticide use and switch to organic farming systems in which synthetic pesticides are replaced by botanical pesticides or chemical control is completely avoided.
Minor concerns:
Ad 1) rcf (or even better x g) does the job. In such situation info on rpm and rotor radius are redundant.
correct typo line 273
– corrected as recommended by the reviewer.
Yours sincerely
Radosław Kowalski

Reviewer 2 Report
The authors adequately addressed my comments and improved their manuscript.
Author Response
Dear reviewer,
Thank you kindly to the reviewer for the review.
Yours sincerely
Radosław Kowalski